# *CYP26A1* Links WNT and Retinoic Acid Signaling: A Target to Differentiate ALDH+ Stem Cells in *APC*-Mutant CRC

**DOI:** 10.3390/cancers16020264

**Published:** 2024-01-07

**Authors:** Caroline O. B. Facey, Victoria O. Hunsu, Chi Zhang, Brian Osmond, Lynn M. Opdenaker, Bruce M. Boman

**Affiliations:** 1Cawley Center for Translational Cancer Research, Helen F. Graham Cancer Center and Research Institute, Newark, DE 19713, USA; caroline.facey@christianacare.org (C.O.B.F.); vhunsu@udel.edu (V.O.H.); johnitch@udel.edu (C.Z.); bosmond@udel.edu (B.O.); lopdenaker@christianacare.org (L.M.O.); 2Department Biological Sciences, University of Delaware, Newark, DE 19716, USA; 3Department Pharmacology & Experimental Therapeutics, Thomas Jefferson University, Philadelphia, PA 19107, USA

**Keywords:** *CYP26A1*, colorectal cancer, cancer stem cells, *APC*, aldehyde dehydrogenase, WNT, retinoic acid, neuroendocrine cells

## Abstract

**Simple Summary:**

This manuscript reports the discovery of a mechanism that links two major cellular pathways, retinoic acid (RA) and WNT signaling, which are involved in colorectal cancer (CRC) development. The link between the two pathways is CYP26A1 enzyme, which controls intracellular RA metabolism and regulates RA signaling. Intracellular levels of CYP26A1 are regulated by WNT signaling, as CYP26A1 is a WNT target gene. Thus, the RA and WNT pathways crosstalk to modulate the metabolism of retinoids by CYP26A1. Mutation of the *APC* tumor suppressor gene drives CRC development by constitutively activating WNT signaling, which increases CYP26A1, its enzymatic degradation of retinoids, and decreases RA signaling. Consequently, when *APC* is mutant, reduced RA signaling leads to incomplete differentiation and overpopulation of ALDH+ stem cells. To restore retinoid-induced differentiation and reduce stem cell numbers, CYP26A1 levels will need to be lowered by inhibiting WNT signaling so that cells can respond to retinoids. Indeed, that is precisely what we observed in our study.

**Abstract:**

*APC* mutation is the main driving mechanism of CRC development and leads to constitutively activated WNT signaling, overpopulation of ALDH+ stem cells (SCs), and incomplete differentiation. We previously reported that retinoic acid (RA) receptors are selectively expressed in ALDH+ SCs, which provides a way to target cancer SCs with retinoids to induce differentiation. *Hypotheses*: A functional link exists between the WNT and RA pathways, and *APC* mutation generates a WNT:RA imbalance that decreases retinoid-induced differentiation and increases ALDH+ SCs. Accordingly, to restore parity in WNT:RA signaling, we induce *wt-APC* expression in *APC*-mutant CRC cells, and we assess the ability of all-trans retinoic acid (ATRA) to induce differentiation. We found that ATRA increased expression of the WNT target gene, *CYP26A1*, and inducing *wt-APC* reduced this expression by 50%. Thus, the RA and WNT pathways crosstalk to modulate CYP26A1, which metabolizes retinoids. Moreover, inducing *wt-APC* augments ATRA-induced cell differentiation by: (i) decreasing cell proliferation; (ii) suppressing *ALDH1A1* expression; (iii) decreasing ALDH+ SCs; and (iv) increasing neuroendocrine cell differentiation. A novel CYP26A1-based network that links WNT and RA signaling was also identified by NanoString profiling/bioinformatics analysis. Furthermore, *CYP26A1* inhibitors sensitized CRC cells to the anti-proliferative effect of drugs that downregulate WNT signaling. Notably, in *wt-APC*-CRCs, decreased CYP26A1 improved patient survival. These findings have strong potential for clinical translation.

## 1. Introduction

For more than half a century, oncologists have had systemic drugs that can induce tumor responses in patients with colorectal cancer (CRC). However, in advanced CRC patients, treatment regimens are almost never curative. Our strategy for addressing this problem has been to determine which cellular mechanisms are responsible for maintaining tissue homeostasis of normal colonic epithelium and, based on this knowledge, determine how dysregulation of these mechanisms gives rise to the development of CRC. This quest has led us to focus on CRC cell biology, the role of the adenomatous polyposis coli (*APC*) gene in colon tumorigenesis, how retinoic acid (RA) and neuroendocrine cell (NEC) signaling controls cellular differentiation, and the mechanisms responsible for the overpopulation of stem cells (SCs) that drives tumor growth (reviewed in [1]).

The primary etiologic factor in colon tumorigenesis is dysregulation of WNT signaling, which occurs in 93% of all CRCs. The *APC* tumor suppressor gene, which functions as a negative regulator of WNT signaling, is the most mutated gene in CRC. The WNT signaling pathway is a multifunctional pathway involved in processes such as stem cell (SC) self-renewal, cellular proliferation, differentiation, and tissue homeostasis. Our discovery that aldehyde dehydrogenase positive (ALDH+) SCs progressively increase in number during CRC development [2] and that this increase coincides with the *APC* zygosity state in hereditary CRC patients led to the concept that *APC* mutation leads to the SC overpopulation that drives CRC development.

Understanding epithelial homeostasis in the colon is also key to identifying how regulatory factors might contribute to tumorigenesis (reviewed in [1]). Tissue renewal in the colonic epithelium depends on the constant self-renewal ability of SCs that reside in the SC niche of the crypts of Lieberkuhn. The extensive proliferation that occurs in the colonic epithelium arises from the SC populations in the crypts. The maintenance of crypt SC populations is controlled by paracrine signaling via neuropeptides secreted from SC-neighboring cells, particularly neuroendocrine cells (NECs). For instance, secretion of glucagon-like peptide 2 (GLP2) and somatostatin from the neighboring NECs controls SC populations and the rate of crypt proliferation [3]. Other factors, such as retinoic acid (RA), that induce SC differentiation along the NEC lineage play a key role in the feedback regulation of crypt SC populations. Thus, defining the growth factors that regulate the differentiation of NECs has implications for our understanding of how incomplete differentiation and SC overpopulation occurs during CRC tumorigenesis.

Our main goal has been to understand how mutation in the *APC* gene causes overpopulation of ALDH+ cancer SCs (CSCs) and decreased differentiation of SCs into neuroendocrine cells (NECs). Since *APC* mutation leads to constitutively activated WNT signaling, this indicates that WNT-signaling plays a key role in controlling SC population size [1,2]. But RA signaling must also play a key role in controlling SC populations since ALDH is a major enzyme in RA signaling and the RA pathway can induce SC differentiation [4,5,6,7]. Thus, we chose to study whether there is a functional connection between WNT and RA signaling, and if *APC* mutation generates an imbalance in a WNT:RA-linked mechanism that contributes to ALDH+ SC overpopulation by impeding retinoid-induced differentiation [8]. We also considered how this mechanism might be targeted in CRC cells.

Previous studies indicate that targeting WNT signaling holds promise because restoring *wild-type APC* expression in *APC*-mutant tissues suppresses tumor growth. For example, inducing *wild-type (wt)-APC* in CRC cells, which contain homozygous mutant *APC*, led to decreased cell proliferation and increased apoptosis [9,10,11]. In mice, inactivation of *Apc* led to intestinal adenoma formation, which provided evidence for crypt SCs as the cells-of-origin of intestinal cancer [12,13]. Restoring *Apc* in murine tumors led to increased enterocyte differentiation, tumor regression, and re-established crypt–villus homeostasis in CRC [13].

Other studies also indicate that targeting RA signaling has potential because retinoids possess anti-cancer activity against CRC tissues. We found that RA receptors are selectively expressed in ALDH+ SCs, which indicates RA signaling mainly occurs in ALDH+ SCs [14]. Since ALDH is a key enzyme in the RA pathway and RA signaling occurs via ALDH+ SCs, this provides a mechanism to selectively target CSCs based on a therapeutic strategy involving retinoid-induced differentiation [15]. Indeed, retinoids have been previously studied as chemo-preventive drugs and systemic therapies in the treatment of CRC and other cancers [16,17]. Furthermore, we found that RA ligands, such as ATRA, induce the differentiation of ALDH+ SCs into NECs [8,18]. On the other hand, *APC* mutation leads to incomplete differentiation in CRC cells [8,19]. Thus, these findings suggest that *APC* mutation attenuates RA signaling in ALDH+ SCs, which contributes to their overpopulation that drives CRC development [8].

Accordingly, to identify mechanisms that functionally link WNT and RA signaling, we investigated how inducing *wt-APC* expression and decreasing WNT signaling affects ATRA’s ability to induce NEC differentiation and reduce ALDH+ CSCs in *APC*-mutant CRC. We also tested drugs that might therapeutically target this mechanism in CRC cells.

## 2. Materials and Methods

### 2.1. Cell Culture and Inducing wt-APC Expression

HT29 colon carcinoma cells containing a ZnCl_2_-inducible *wt-APC* or β-galactosidase (control) expression vector were provided by Dr. Bert Vogelstein’s lab (Johns Hopkins University) [9]. We refer to the modified HT29 cells as HT29-APC or HT29-GAL cells. HT29 cells were chosen for most of the study because they contain a homozygous *APC* mutation and do not have any known retinoic acid receptor gene mutations. Modified cells were cultivated in McCoy’s 5A modified media (Gibco/Thermo Fisher Scientific, Waltham, MA USA) supplemented with 10% fetal bovine serum (FBS; Atlanta Biologicals/R&D Systems, Minneapolis, MN, USA), 1% penicillin:streptomycin (pen-strep; VWR), and 600 μg/mL hygromycin B (Thermo Fisher Scientific, Waltham, MA, USA). All variations of HT29 cells grew equally well in terms of cell density. Respective expression plasmids were induced with 120 μM ZnCl_2_ (Sigma-Aldrich, St. Louis, MO, USA) for the times indicated. HT29-APC cells treated with ZnCl_2_ are referred to as *wt-APC* and those not treated with ZnCl_2_ are referred to as mt-*APC*. All cells were maintained in a 37 °C incubator with 5% CO_2_.

### 2.2. Cell Proliferation

HT29 cells were plated at 12,000 cells per well into 96-well plates. Cells were allowed to adhere for 24 h and serum-starved for an additional 24 h. Cells were then treated for 48 h with ZnCl_2_ and/or ATRA (Stem Cell Technologies, Vancouver, BC, Canada), as indicated. Controls involved no ZnCl_2_ or DMSO (0.1%; Fisher Scientific, Waltham, MA, USA) vehicle. After the indicated time, cells were washed in PBS and stained in 0.5% crystal violet (Sigma-Aldrich, St. Louis, MO, USA) solution for 20 min. Excess crystal violet was then washed off with deionized water and left to dry at room temperature for at least 24 h. Dried crystal violet stain was then re-suspended in 10% acetic acid solution and the optical density measured at 570 nm using a Tecan Infinite F200 Pro Multi-Plate Reader. Data were quantified using Tecan i-Control 1.8 software and analyzed using Microsoft Excel 365. Values reflect cell proliferation, as our standard curve showed that the absorbance values from crystal violet stain were directly proportional to cell number. The proliferation index was calculated by normalizing cell proliferation index of treated to untreated cells. Biological replicates were performed in triplicate with 6 technical replicates per condition, and the average was plotted with standard error of the mean (SEM) indicated.

### 2.3. WNT/β-Catenin Activity

WNT/β-catenin activity was measured using the TCF/LEF Reporter (BPS Biosciences, San Diego, CA, USA) and Two-Step or Dual Glo (Firefly and Renilla) Luciferase Assays (BPS Biosciences or Promega, Madison, WI, USA) according to the manufacturer’s instructions. In brief, cells were plated onto clear-bottom, white, 96-well plates and adhered for 24 h. Next, cells were transfected with the TCF/LEF reporter plasmids using Lipofectamine 3000 (Thermo Fisher Scientific, Waltham, MA, USA). Cells were then treated as indicated for 24 h. The firefly and renilla luciferase assays were performed, luminescence was recorded using a Tecan Multi-Plate Reader with i-Control 1.8 software, and data were analyzed using Microsoft Excel 365. Luciferase activity was measured by calculating the ratio of firefly to renilla luminescence signals. Biological replicates were performed in triplicate with 6 technical replicates per condition. The average luciferase activity was plotted with SEM indicated.

### 2.4. NanoString Profiling

To measure the effects of manipulating WNT and RA signaling, we utilized the nCounter PanCancer Pathways Panel from NanoString (Seattle, WA, USA) to provide high-throughput data, including differential gene expression and subsequent pathway analysis. The PanCancer Pathways Panel includes 770 genes involved in several cancer pathways, including WNT, hedgehog, apoptosis, cell cycle, RAS, PI3K, STAT, MAPK, Notch, TGF-β, chromatin modification, transcriptional regulation, and DNA damage control [20], and we included an additional 55 genes of interest to evaluate SC markers, differentiation markers, ALDH isoforms, RA receptors, and HOX genes.

HT29-*APC* cells were plated at 700,000 cells per well onto 6-well plates and allowed to adhere for 24 h. Cells were then serum-starved for 24 h. Cells were then treated with 15 µM ATRA or 0.1% DMSO and 120 µM ZnCl_2_ or no ZnCl_2_ for 24 h. Total RNA was isolated from cell lysates using the Trizol method. Samples were frozen at −80 °C and sent to Wistar Institute’s Genomic Core Facility to analyze the quality of the RNA and subsequent NanoString profiling using the PanCancer Panel. Results were obtained in RCC file format and data were analyzed using NanoString nSolver 4.0 software [20]. A heatmap was then generated to show differential expression of 785 mRNAs amongst the four treatment conditions (+ZnCl_2_ & +ATRA, +ZnCl_2_ & −ATRA, −ZnCl_2_ & +ATRA, −ZnCl_2_ & −ATRA). Ratios were determined using nSolver 4.0 software, significance was assessed, and results were plotted in Microsoft Excel 365. Gene lists of interest were analyzed using the STRING database for protein–protein interaction networks and functional enrichment [21].

### 2.5. Western Blotting and Densitometry

Standard Western blotting procedures were performed. Cells were plated, treated as indicated with 120 µM ZnCl2 (Millipore Sigma, St. Louis, MO, USA), ATRA, or DMSO, and lysed with RIPA (Millipore Sigma) containing Halt Protease and Phosphatase Inhibitor (Thermo Fisher, Waltham, MA, USA) and cysteine protease inhibitor E-64 (Millipore Sigma). Cell lysates were scraped from dishes, centrifuged at 12,000 rpm for 15 min, and stored at −20 °C. Protein concentrations were determined using the Pierce BCA Protein Assay Kit (Thermo Fisher) and 100 µg total protein was prepared in Laemmli buffer (Bio-Rad, Hercules, CA, USA) and 2.5% or 5% β-mercaptoethanol (Bio-Rad). Samples were loaded onto 4–20% mini-precast gradient gels (Bio-Rad) for SDS-PAGE at 200 V for 45 min. Proteins were then transferred to PVDF membranes (Thermo Fisher) at 90 V for 2 h. Blots were then blocked in 5% BLOTTO (nonfat dried milk in tris-buffered saline [TBS] with 0.1% Tween 20 [TBST]) for 1 h. Next, blots were incubated in primary antibody prepared in BLOTTO overnight on a rocking platform at 4 °C. Blots were then washed in TBST 3× for 10 min. Secondary antibody was prepared in BLOTTO and blots were incubated for 1–2 h at room temperature. Specific antibodies are listed in Appendix A. Blots were then washed in TBST 3× for 10 min and incubated in Super Signal West Dura Extended Duration Substrate (Thermo Fisher) for 5 min to detect horseradish peroxidase conjugation. Blots were then imaged using the LiCor Odyssey Fc developer and Image Studio 5.8 software (model number 2800, LI-COR Biosciences, Lincoln, NE, USA). Densitometry was performed using Image Studio and graphs were plotted using Microsoft Excel 365. Experiments were performed 3 times, standard error of the mean (SEM) was calculated, and graphs were plotted as shown.

### 2.6. Flow Cytometry and Fluorescence Activated Cell Sorting 

NEC and ALDH+ (ALDEFLUOR+) cells were quantified and/or isolated using GLP2R antibody conjugated to Alexafluor 594 (Novus Biologicals, Centennial, CO, USA) with IgG control (R&D) or ALDEFLUOR Assay (Stem Cell Technologies, Vancouver, BC, USA) according to the manufacturer’s instructions. Flow cytometry was performed on a BD FACS Aria II and FACS was performed on a BD LSR Fortessa. Data were processed using FACS Diva 9.0 software and graphs were plotted in Microsoft Excel 365 with error bars indicating SEM. Experiments were performed 3 times. 

### 2.7. Determining Synergistic, Additive, or Antagonistic Anti-Proliferative Effects of CYP26A1 and WNT Signaling Inhibitors

We investigated the anti-proliferative effects of the *CYP26A1* inhibitors Liarozole and Talarozole in combination with agents (Sulindac or Piroxicam) that have anti-tumor activity against *APC*-mutant tissues (reviewed in [16]). HT29 and HCT116 cells were cultivated in McCoy’s 5A media supplemented with 10% FBS and 1% pen-strep. SW480 cells were cultivated in Leibovitz L15 media supplemented with 10% FBS and 1% pen-strep. Cells were plated, allowed to adhere for 24 h, and serum-starved for 24 h. The cells were then treated for 48 h with the various agents as single drugs or in combination. For *CYP26A1* inhibitors, the IC50 doses of Liarozole and Talarozole were respectively as follows: HT29 cells, 80 and 30 µM; HCT116 cells, 110 and 22 µM; and SW480 cells, 80 and 25 µM. For WNT inhibitors, the IC50 doses of Sulindac and Piroxicam were respectively as follows: HT29 cells, 680 µM and 550 µM; HCT116 cells, 450 µM and 600 µM; SW480 cells, 220 µM and 270 µM. In dose–response experiments, Sulindac and Piroxicam doses were varied based on the IC50 dose (75% of IC50, IC50, and 125% of IC50). For clarity, all concentrations are listed in Appendix A. The proliferation index was determined using the crystal violet assay (see above). Synergistic, additive, or antagonistic effects were calculated based on the Bliss independence dose–response model [22]. For example, these effects were calculated as the ratio of the observed response (observed values for the combination of drugs a and b) divided by the predicted response (value for individual drug a multiplied by individual value for drug b), based on the Bliss independence dose–response model. Further analyses were performed as described in the main and Appendix A legends.

### 2.8. Patient Survival Studies

RNA-seq, genotype, and clinical data of CRC patients were obtained from the Genomic Data Commons (GDC) data portal https://portal.gdc.cancer.gov/ (accessed on 15 May 2023). For individual patients, records of their *APC* genotypes, survival data, as well as *CYP26A1* expression in both human CRC and normal human colon samples were paired together. These patients were then grouped based on whether they carried *APC* mutations and the change (either increase or decrease) in *CYP26A1* expression between CRC and normal colon tissue samples. 

### 2.9. Statistical Analysis

Statistics were calculated either in Microsoft Excel 365 using the *t*-test function or GraphPad Prism 10 using one-way ANOVA, two-way ANOVA, or multiple unpaired *t*-test, as indicated in the figure legends. For patient survival studies, Kaplan–Meier survival curves of different groups were compared using log-rank tests. Statistical significance was based on *p* < 0.05.

## 3. Results

### 3.1. Inducing wt-APC Decreases WNT Signaling and Reduces Expression of WNT Target Genes

We first wanted to show that the HT29-*APC* cell line established by Morin et al. [9] was functioning correctly in the ability of ZnCl_2_ treatment to induce *wt-APC* expression. Using two independent experimental approaches, we confirmed that expression of *wt-APC* occurred in response to ZnCl_2_ treatment (Appendix A). We initially measured WNT activity utilizing the TopFlash luciferase reporter assay, which measures TCF/LEF transcriptional activity. There was a 90% reduction in WNT activity in ZnCl_2_-treated cells compared to untreated cells (Appendix A). We then confirmed that reduced expression occurred in several proteins encoded by WNT/ β-catenin target genes [23], including *MET* [24], *JUN* [25], *CD44* [26], and *MYC* [27,28] (Appendix A).

### 3.2. ATRA Promotes WNT/β-Catenin Activity, wt-APC Attenuates ATRA’s Effect

Although ATRA reduces the number of ALDH+ SCs, reduces sphere-forming ability, and promotes differentiation of SCs in CRC [14], here we found that total cell proliferation in cells was reduced less than 20% at 48 h when treated with 20 µM ATRA (Figure 1A). However, when *wt-APC* expression was induced in HT29 cells, ATRA led to a further reduction in total cell proliferation by 42% compared to untreated controls (Figure 1A). Dose–response analysis also showed that cells with *wt-APC* were 2.4 times more responsive to the anti-proliferative effects of ATRA than cells with mt-*APC* (based on line slopes; Figure 1B). Additionally, we observed that WNT/β-catenin activity was increased by ATRA in HT29 cells by up to 10 times as compared to controls (Figure 1C); but, this increase in WNT activity was significantly attenuated with the expression of *wt-APC* (Figure 1C). 

### 3.3. Inducing wt-APC Decreases ALDH+ Stem Cells and Increases NEC Differentiation

Here, we show an increase in the percentage of GLP2R+ NECs when *wt-APC* expression was induced (Figure 2A). The relative number of GLP2R+ cells quadrupled when *wt-APC* was induced and increased 6-fold with combined *wt-APC* expression and ATRA as compared to mt-*APC* and no ATRA (Figure 2B). In parallel, the percentage of ALDH+ (ALDEFLUOR+) cells decreased by approximately 20% with ATRA alone and almost 60% with the combination of ATRA treatment and *wt-APC* induction (Figure 2C). In a similar manner, mRNA expression of *ALDH1A1* decreased by 5-fold with the ATRA and *wt-APC* combination as compared to controls (Figure 2D). These results further supported the existence of interplay between the WNT and RA signaling pathways.

We also investigated whether inducing *wt-APC* in HT29-*APC* cells changed the expression of NEC differentiation cell markers. Indeed, several NEC markers increased in expression with the expression of *wt-APC* in HT29 cells. Increased protein expression of CHGA, GLP2R, NSE (ENO2), and SSTR1 occurred, as shown in Figure 3. Overall, we found that inducing *wt-APC* decreased ALDH+ CSCs and increased NEC differentiation.

### 3.4. Expression Profiling Identified CYP26A1 as a Link between WNT and RA Signaling

NanoString profiling revealed differential expression of 785 mRNAs across four treatment groups from the results of ZnCl_2_ treatment to induce *wt-APC* expression and ATRA treatment to increase RA signaling (+ZnCl_2_ & +ATRA, +ZnCl_2_ & −ATRA, −ZnCl_2_ & +ATRA, −ZnCl_2_ & −ATRA). Hierarchical, agglomerative clustering was achieved and shown as a heatmap, which was generated using nSolver 4.0 software (Figure 4A).

To examine the effect of inducing *wt-APC* expression and increasing RA signaling at the same time, we calculated the ratio of treated cells (+ZnCl_2_ & +ATRA) to the corresponding control group (−ZnCl_2_ & −ATRA) using nSolver 4.0 software. A set of 248 genes was identified as being differentially expressed (Appendix A). To identify predicted signaling pathway interactions, we analyzed the set of 248 genes using the STRING database [21]. The large number of predicted interactions found in this network analysis (Appendix A) suggested that a substantial level of crosstalk occurs when the WNT and RA signaling pathways are modulated in HT29 cells.

By analyzing the list of 248 genes using the DAVID functional tool [29] and ranking the list of over 1500 hits, we identified 14 genes that are involved in RA signaling using the GOTERM_BP_DIRECT list. We then re-analyzed this list of 14 genes using the STRING database and found several predicted interactions. Of the 14 genes listed as involved in RA signaling, 12 were identified to have protein interactions: CYP26A1, RARA, HDAC2, KLF4, TNF, LEP, PDGFA, MYB, DKK1, WNT7B, WNT11, and FZD7 (Figure 4B). Notably, five WNT target genes were identified: CYP26A1 [30], WNT7B, WNT11, FZD7, and DKK1 (listed on NanoString’s Gene to Pathway Summary). However, CDKN2D and HOXA2 did not show any predicted connections with the other proteins in the STRING network.

The highest change in expression occurred in *CYP26A1*, with a 15.52-fold increase when both *wt-APC* expression is induced and cells are treated with ATRA (Figure 4C). Moreover, HT29 cells treated with ATRA alone experience a 30.8-fold increase in *CYP26A1* expression and inducing *wt-APC* attenuated this increase by 50% (Figure 5A). Western blot experiments validated the 50% decrease in *CYP26A1* expression when cells were induced to express *wt-APC* and treated with 20 µM ATRA (Figure 5B,C). 

### 3.5. CYP26A1 Inhibitor Agents Sensitize CRC Cells to the Anti-Proliferative Effect of Drugs That Downregulate WNT Signaling

The anti-proliferative effects of CYP26A1 inhibitors Liarozole and Talarozole were evaluated alone and in combination with Sulindac and Piroxicam, which have anti-tumor activity against *APC*-mutant tissues (Figure 6 and Appendix A). Three CRC cell lines with mutations that activate WNT signaling were used: HT29 and SW480 have mutant *APC* and HCT116 with mutant *CTNNB1*. Each line has a different RA receptor genotype. HT29 cells have wild-type retinoid receptors, SW480 cells have mutations in *RARA* and *RXRG*, and HCT116 cells have mutant *RARA* [16]. Given that Sulindac and Piroxicam have efficacy against *APC*-mutant tissues, they likely act to downregulate WNT signaling. Indeed, Sulindac is known to target β-catenin and downregulate WNT signaling (reviewed in [16]).

CRC cells were treated with the various agents as single drugs or in combination (Figure 6 and Appendix A). The Sulindac and Piroxicam doses were varied based on the IC50 (half-maximal inhibitory concentration) dose for treated cell lines. The data revealed that anti-tumor agents Sulindac and Piroxicam decreased the proliferation of CRC cells. The data also showed that CYP26A1 inhibitors Liarozole and Talarozole alone reduced CRC proliferation. Moreover, when Sulindac or Piroxicam was combined with Liarozole or Talarozole, the drug combination produced additive or synergistic effects (Figure 6C,F). Further dose–response analysis of the anti-proliferative effects of varying Piroxicam and Sulindac concentrations showed that the Liarozole + Piroxicam, Liarozole + Sulindac, and Talarozole + Sulindac combinations produced the highest dose responsiveness in HT29, SW480, and HCT116 cells, respectively (Appendix A).

### 3.6. Analysis of Human CRC Cases Indicates CYP26A1 Predicts Survival of Patients with Wild-type APC Tumors

To evaluate the effects of *CYP26A1* on CRC patient survival, we conducted multi-omics bioinformatics analysis using RNA-seq, mutation, and clinical data obtained from Genomic Data Commons (GDC). Patients were grouped based on their *APC* genotypes and changes (increase or decrease) in *CYP26A1* expression level between CRC and normal human colon samples. We found that the majority of CRC patients expressed increased levels of *CYP26A1* in their CRC samples compared to normal colon tissue from the same patient, and this difference was statistically significant (Figure 7A). By itself, this change in *CYP26A1* expression level was not a statistically significant predictor of patient survival (Figure 7B). However, when the genotype of *APC* was used to further sub-group the patients, we found that *CYP26A1* expression was a significant predictor of patient survival only when patient tumors were *wild-type APC* (Figure 7C). But, estimated survival was not statistically significant between increased and decreased *CYP26A1* expression groups among those patient tumors carrying *APC* mutations (Figure 7D). Similarly, when patients were sub-grouped based on *CYP26A1* level, *APC* mutation was not a significant predictor of patient survival (Appendix A).

## 4. Discussion

### 4.1. Main Findings of Our Study

We previously established that overpopulation of ALDH+ CSCs correlates with the zygosity state of *APC* mutation during the stepwise tumor development in FAP patients [1]. We also found that *APC* mutation causes failure of ALDH+ SCs to mature into NECs [8,18]. Together, these findings provide a clue to a functional interaction between the WNT and RA signaling pathways, which maintain stemness and promote differentiation, respectively [8,14,18]. Accordingly, we conjectured that a link exists between the WNT and RA pathways, and, when *APC* is mutant, an imbalance in a WNT:RA-linked mechanism promotes CRC development [8]. Our main finding herein identified *CYP26A1* as a link between WNT and RA signaling that can be targeted to decrease ALDH+ SCs and increase retinoid-induced differentiation of *APC*-mutant CRC cells. The functional relevance of this finding is that (1) *CYP26A1* expression is controlled by WNT signaling via *APC* [30], and (2) *CYP26A1* enzyme controls intracellular RA metabolism and regulates RA signaling [16,17]. Our previous finding that RA signaling mainly occurs via ALDH+ SCs [14] raises two key questions addressed below:
***Q1*** *How is RA signaling regulated in ALDH+ SCs?*
***Q2*** *How does dysregulation of RA signaling due to APC mutation contribute to the overpopulation of ALDH+ SCs that drives the development of CRC?*


### 4.2. Studying the Effect of wt-APC on ATRA Response Indicates That WNT Signaling, via Its Target Gene CYP26A1, Regulates RA Signaling in the Differentiation of ALDH+ SCs

We found that inducing *wt-APC* expression, which decreases WNT signaling, reduced the expression of several WNT target genes and increased the sensitivity of HT29 cells to the anti-proliferative effects of ATRA (Figure 1 and Appendix A). We also found that ATRA treatment of HT29 cells significantly increased *CYP26A1* expression (30.8-fold) and inducing *wt-APC* attenuated this increase by 50% (Figure 5A). This increase in *CYP26A1* by ATRA treatment of *APC*-mutant HT29 cells may be explained by two factors: (1) *CYP26A1* is a RA target gene [31,32,33], which should be induced by ATRA; and (2) *CYP26A1* is also a WNT target gene [30] and ATRA increases WNT/β-catenin activity in HT29 cells even though they already have constitutively activated WNT due to mutated *APC* (Figure 1C). Notably, increased *CYP26A1* expression in ATRA-treated HT29 cells decreased when they were induced to express *wt*-*APC* (Figure 5). This finding indicates that the ability of *wt*-*APC* to reduce WNT/β-catenin activity is the main mechanism that counterbalances the ability of ATRA to increase CYP26A1 expression. Consequently, since *wt-APC* inhibits WNT signaling, it should maintain the balance between WNT and RA signaling. Indeed, that is what we found because induction of *wt-APC* reversed ATRA’s ability to increase: (*i*) WNT/β-catenin activity (Figure 1C) and (*ii*) *CYP26A1* expression (Figure 5). These findings show that modulation of *CYP26A1* levels by WNT signaling regulates RA signaling in *APC*-mutant cells. Thus, our findings indicate that the anti-proliferative effect of ATRA is greatly enhanced by decreased expression of CYP26A1 due to *wt*-*APC’s* ability to reduce WNT signaling.

Since the RA pathway components are mainly expressed in ALDH+ SCs [14], WNT signaling, via its target gene *CYP26A1*, likely regulates RA signaling in ALDH+ SCs. Indeed, inducing *wt-APC* expression reduced *CYP26A1* expression and enhanced the ability of ATRA to decrease CRC cell proliferation (Figure 1 and Figure 5). Moreover, inducing *wt-APC*: (*i*) suppressed ALDH1A1 expression, (*ii*) decreased ALDH+ SCs, and (*iii*) increased neuroendocrine cell differentiation. These findings show that *APC’s* role in the control of WNT signaling regulates, via its target gene *CYP26A1*, RA signaling in the differentiation of ALDH+ CSCs.

To further explore the mechanism by which WNT signaling might regulate RA signaling, we examined differential mRNA expression in HT29 cells that were induced to express *wt-APC* and were treated with ATRA compared to their respective controls. NanoString profiling revealed that among the 248 genes showing a significant change in expression, *CYP26A1* showed the greatest increase in expression (Figure 4C and Appendix A). Additionally, our NanoString and bioinformatics analyses identified a novel CYP26A1-based network of protein interactions that revealed how the WNT and RA pathways are interconnected. Specifically, we found a unique signaling cascade that links components of the RA signaling pathway (CYP26A1, RARA) with components of the WNT pathway (DKK1, WNT7B, WNT11, FZD7) via HDAC2, KLF4, and TNF (Figure 4B). Thus, a CYP26A1-based network of both RA and WNT signaling components is predicted to play a role in the functional link between these two pathways.

### 4.3. Studying the Effect of APC Mutation on NEC Differentiation Indicates That Decreased RA Signaling Contributes to the Overpopulation of ALDH+ SCs That Drives the Development of CRC

We found that inducing *wt-APC* expression, which decreased WNT-signaling and lowered *CYP26A1* expression, led to a decrease in cell proliferation (Figure 1A,B) and an increase in NEC differentiation (Figure 2A,B and Figure 3). Inducing *wt-APC* also decreased ALDH+ SC numbers and reduced the expression of the ALDH1A1 SC marker (Figure 2C,D). Altogether, these findings show that elevated *CYP26A1* levels prevent differentiation of CSCs in *APC*-mutant cells by increasing RA clearance, which reduces RA signaling (Figure 8). This mechanism provides an explanation for how decreased RA signaling contributes to the ALDH+ SC overpopulation that drives CRC development.

We also showed that inducing *wt-APC* expression in CRC cells sensitizes them to ATRA’s ability to promote NEC differentiation. This increase in NEC differentiation likely occurs because of interplay between the WNT and RA signaling pathways. In this view, the induction of *wt-APC* decreases WNT signaling, which lowers the level of stemness sensitizes cells to the effects of ATRA treatment by increasing RA signaling, which promotes cellular differentiation. Moreover, the differentiation of SCs along the NEC lineage is augmented because RA signaling mainly occurs in ALDH+ SCs which will have an increased rate of differentiation when their stemness is decreased. In other words, NECs were formerly ALDH+ SCs that underwent differentiation in response to RA signaling. Our previous studies also indicate that NEC signaling, via neuropeptide secretion, auto-regulates the rate of NEC maturation in a feedback manner as SCs mature along the NEC lineage, which can contribute to control of SC proliferation [8,14,18]. Thus, our data indicate that RA signaling is fully functional in ALDH+ SCs when *APC* is wild-type so that the induction of RA signaling can inhibit the growth of ALDH+ SCs by promoting their differentiation along the NEC lineage. During tumorigenesis, *APC* mutation will have the opposite effect—activated WNT will increase stemness, which will decrease the rate of CSC differentiation into NECs and contribute to overpopulation of ALDH+ CSCs. Given that CYP26A1 links RA and WT signaling, we surmise that it is aberrant regulation of *CYP26A1* by both WNT and RA signaling that underlies these differences between normal and CRC tissue dynamics.

Other studies have shown that changes in *CYP26A1* expression are associated with an altered tissue phenotype. For example, Shelton et al. [30] found that *CYP26A1* was upregulated in the intestine of *Apc*-mutant zebrafish embryos (and in *Apc*^Min/+^ mouse adenomas, human FAP adenomas, and human sporadic CRCs), but not in zebrafish embryos with *wt-Apc*. Defects in gut differentiation were also reversed with pharmacologic inhibition or knockdown of *CYP26A1* in *Apc*-mutant zebrafish embryos [30]. Further studies on zebrafish showed that *Apc* has a dual role in regulating Wnt and RA signaling [34,35]. Therefore, expression of *wt-APC* or pharmacologic inhibition of *CYP26A1* may provide a novel therapeutic strategy to sensitize CRC cells to ATRA [36]. Indeed, a recent study by Penny et al. involved treating *Apc*^Min/+^ mice with the *CYP26A1* inhibitor Liarozole [37]. The administration of Liarozole to *Apc*^Min/+^ mice increased endogenous RA signaling (by blocking ATRA metabolism) and dramatically reduced intestinal adenoma numbers in *Apc*-mutant mice. We also found that treatment of human CRC cells with Liarozole decreased proliferation, sphere formation, and the size of the ALDH+ SC population [14]. This suggests that decreasing the intracellular metabolism of ATRA using agents that inhibit CYP26A1 activity may increase ATRA levels and therapeutically augment RA signaling, leading to decreased CSC numbers in *APC*-mutant CRC tissues [16].

### 4.4. Clinical Significance of Our Results That Show CYP26A1 Inhibitors, Which Block RA Metabolism, Sensitize CRC Cells to the Anti-Proliferative Effect of Drugs That Downregulate WNT Signaling

Our findings on testing CYP26A1 inhibitors demonstrate their potential clinical importance in CRC. For instance, given that RA receptor-mutant CRC cells showed a similar response to *CYP26A1* inhibitors (Liarozole and Talarozole) as non-mutant cells (Figure 6), this finding indicates that the inhibition of *CYP26A1* leads to a high enough intracellular RA level to induce growth inhibition regardless of RA receptor genotype. We then investigated the effects of these *CYP26A1* inhibitors in combination with agents (Sulindac or Piroxicam) that have anti-tumor activity against *APC*-mutant tissues. As noted above, Sulindac has been shown to target β-catenin and downregulate WNT signaling (reviewed in [16]). Notably, when Sulindac or Piroxicam was combined with *CYP26A1* inhibitors Liarozole or Talarozole, the drug combination produced additive or synergistic effects (Figure 6C,F). 

That the drug combinations reduce CRC proliferation to a greater extent than the *CYP26A1* inhibitors or anti-WNT agents alone indicates that inhibiting *CYP26A1* to increase RA signaling combined with the effect of inhibiting WNT signaling holds great promise as a therapeutic approach in oncology. Certainly, there are emerging roles for WNT inhibitors as well as retinoids and retinoic acid metabolism blocking agents in the treatment of cancers [16,17].

### 4.5. Study of Human CRC Cases Demonstrates That CYP26A1 Predicts Patient Survival According to APC Genotype

The bioinformatics analysis of human CRC cases provided further evidence that *CYP26A1* has clinical significance (Figure 7). We found that the majority of CRC patients expressed an increased level of *CYP26A1* in their tumor tissues compared to normal colon tissue from the same patient. Considering that *CYP26A1* is a key enzyme responsible for RA degradation, patients who express lower levels of *CYP26A1* should have higher levels of RA in their tumor cells, which should upregulate RA signaling activity and endow CRC patients with better survival. This prediction was supported by our results showing that *CYP26A1* was only a significant determinant of patient survival when patient tumors carried wild-type, but not mutant, *APC*. This finding also supported the existence of crosstalk between WNT and RA signaling that is dependent upon the *CYP26A1* expression level (hence cellular RA level) and *APC* genotype of the patients.

### 4.6. Relations and Roles of CYP26A1 in the Context of Decreased RA Signaling and Increased ALDH+ SCs in APC-Mutant CRCs

Given the pivotal role that CYP26A1 plays in controlling retinoid metabolism and RA signaling, our understanding of how dysregulated CYP26A1 expression contributes to tumorigenesis should help us to develop new CYP26A1-based therapeutic strategies to target CSCs in CRC. Indeed, in human CRCs, CYP26A1 is significantly overexpressed in tumor tissues compared to normal colon tissues [30,38,39,40]. As noted above, elevated levels of *CYP26A1* can be attributed to the fact that *CYP26A1* is a WNT target gene and most CRCs have activated WNT signaling due to mutant *APC*. Thus, upregulation of *CYP26A1* should lead to increased RA clearance and decreased RA signaling, which contribute to the incomplete differentiation that is a hallmark of CRC. In this line of reasoning, an impaired ability to differentiate provides an explanation for how overpopulation of ALDH+ SCs occurs in CRC, because previous studies show that RA signaling mainly occurs in ALDH+ cells [14]. Thus, it can be deduced that to therapeutically restore retinoid-induced differentiation and reduce SC numbers, CYP21A1 levels will need to be lowered by inhibiting WNT signaling so that cells become responsive to the anti-proliferative and differentiation-inducing effects of retinoids. Indeed, that is precisely what we observed in our study. Overall, our findings indicate that targeting CYP26A1 represents a promising approach to enhance retinoid-induced differentiation of ALDH+ CSCs.

## 5. Conclusions

The clinical significance of our study is based on showing how CRC CSCs might be sensitized to retinoid-induced cellular differentiation. We present a mechanism and therapeutic strategy to explain how differentiation-inducing agents such as ATRA may be used to promote CSC differentiation. Indeed, RA is an effective therapeutic agent in the treatment of acute promyelocytic leukemia (APL) [41,42,43,44,45]. RA therapy has also been shown to improve the survival of neuroblastoma patients [46]. While using retinoids to treat other cancers has shown limited success [16,17], therapeutically inhibiting CYP26A1, as we have shown herein, to reduce RA metabolism may prove useful in designing approaches to promote the differentiation of CSCs with retinoids. Therefore, our findings are important because CRC development is driven by the overpopulation of CSCs, which are cell populations that show resistance to immunotherapy as well as conventional chemotherapy and radiation [47,48,49,50,51,52,53,54,55,56,57,58]. Thus, finding ways to promote retinoid-induced differentiation of CSCs in vivo may lead to new and more effective therapeutic strategies for CRC patients.

## Figures and Tables

**Figure 1 cancers-16-00264-f001:**
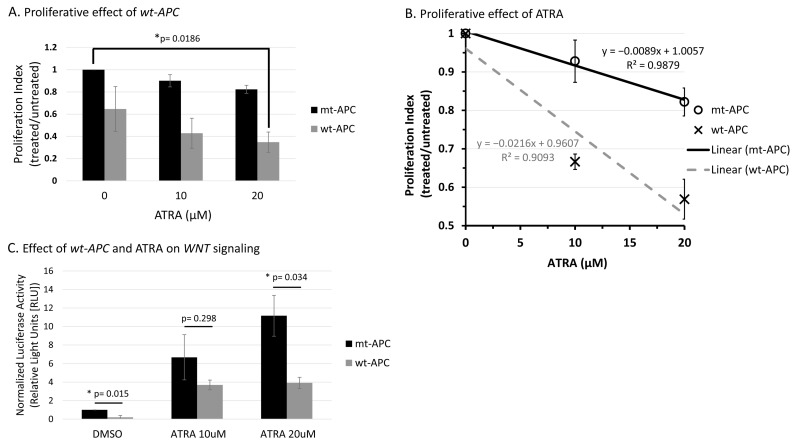
*Wt-APC* expression sensitizes HT29 cells to ATRA’s ability to decrease cell proliferation and attenuates ATRA-induced WNT/β-catenin activity. (**A**) Cell proliferation was reduced by 65% when *wt-APC* was induced by ZnCl_2_ and cells were treated with 20 µM ATRA for 48 h. Values are normalized to *mt-APC* DMSO control. (**B**) Cell proliferation was reduced by approximately 10% (10 μM) and 20% (20 μM) in mutant-*APC* (*mt-APC*) cells (black line). Cell proliferation was further reduced by approximately 36% (10 μM) and 42% (20 μM) in wild-type *APC* (*wt-APC*) cells (dashed gray line). Similar results were seen at 24 h, altogether indicating a time- and concentration- dependent effect. *Mt-APC* and *wt-APC* values were normalized to each of their respective DMSO controls. (**C**) TCF/LEF reporter assay was performed to determine WNT/β-catenin activity. ATRA caused an increase in WNT activity in mt-*APC* cells (black bars). However, this increase was partially attenuated when *wt-APC* expression was induced (gray bars). Experiments (*n* = 3) were performed with error bars plotted as standard error of the mean (SEM). Multiple unpaired *t*-test was used to determine statistical significance at *p* < 0.05.

**Figure 2 cancers-16-00264-f002:**
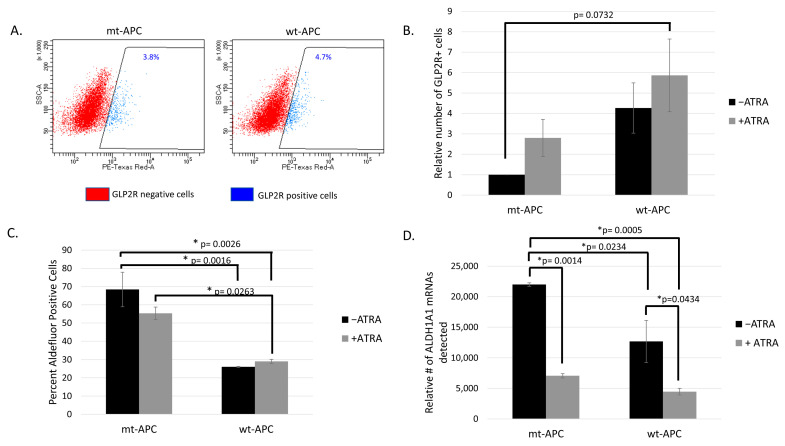
Inducing *wt-APC* enhances differentiation of neuroendocrine cells: GLP2R+ cells increase and ALDH+ cells decrease. (**A**) GLP2R+ and GLP2R− cells were sorted from parent HT29 cells (mt-*APC*) and cells induced to express *wt-APC*. The percentage of GLP2R+ subpopulation is shown. (**B**) Cells were stained for GLP2R, and relative number was calculated by normalizing to untreated mt-*APC* cells. The relative number of GLP2R+ cells quadrupled when *wt-APC* was induced (black bars) and increased by 6-fold with the combination of *wt-APC* and ATRA (comparing mt-*APC* black bar and *wt-APC* gray bar). (**C**) The percentage of ALDH+ SCs cells decreased by ~20% with ATRA alone and up to ~60% when HT29 cells were induced to express *wt*-*APC*. (**D**) Results from NanoString profiling show a similar decrease in *ALDH1A1* mRNA in response to ATRA and *wt-APC*. Experiments were performed 3 times and statistical significance was determined by two-way ANOVA with multiple comparisons.

**Figure 3 cancers-16-00264-f003:**
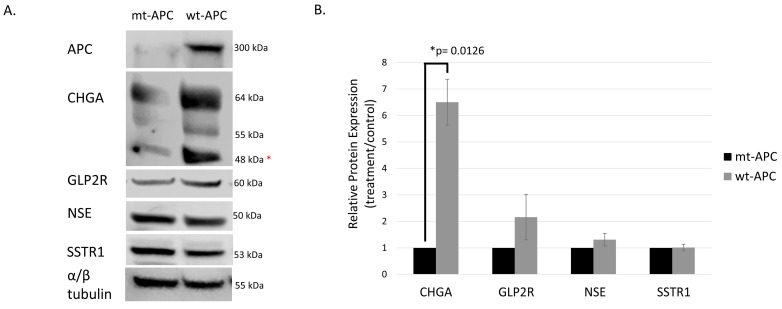
Protein expression of neuroendocrine markers is increased with induction of *wt-APC*. HT29 cells were induced to express *wt-APC*. Western blotting (**A**) and densitometry (**B**) were performed to analyze the protein expression of neuroendocrine markers. Expression of full-length *APC* is shown and verified in the top right panel of (**A**). Chromogranin A (CHGA), GLP2R, and NSE/ enolase were increased with *wt-APC,* but SSTR1 was increased to a lesser extent. * The lower CHGA band was analyzed for densitometry, higher bands may indicate post-translational modification such as glycosylation of the protein. The experiment (*n* = 3) was performed with averages plotted in (**B**) and a representative blot is shown in (**A**). Statistical significance was determined by multiple unpaired *t*-test. The original whole blot of Figure 3A is included in Appendix A- Western Blots.

**Figure 4 cancers-16-00264-f004:**
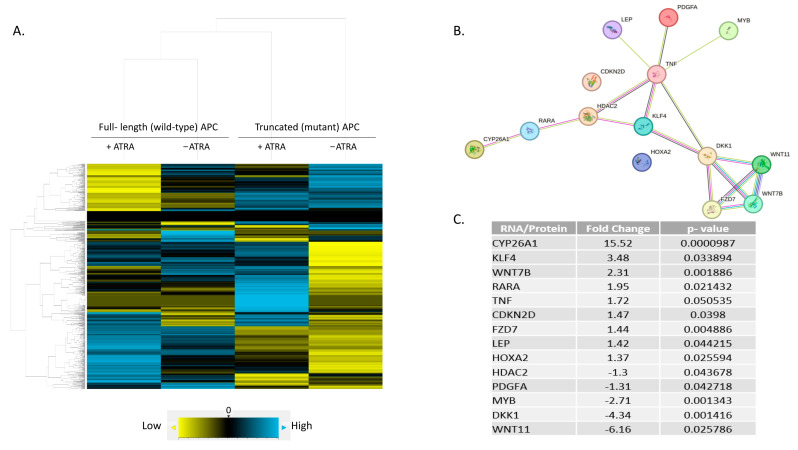
A predicted protein network linking RA and WNT pathways. (**A**) Hierarchical, agglomerative clustering using NanoString nSolver 4.0 software generated a heatmap showing differential expression of 785 mRNAs amongst four treatment conditions. Yellow = low expression, blue = high expression, black = negligible expression at or below background level. (**B**) A subset of 248 genes was identified with significant fold-change in expression between *wt-APC* + ATRA versus mt-*APC* − ATRA (Appendix A). DAVID analysis identified a list of 14 genes involved in cellular response to RA (**C**). STRING database analysis of the 14 genes identified network interactions shown in (**B**). https://David.Ncifcrf.Gov/ accessed on 27 August 2022 and https://string-db.org/ accessed on 1 August 2023.

**Figure 5 cancers-16-00264-f005:**
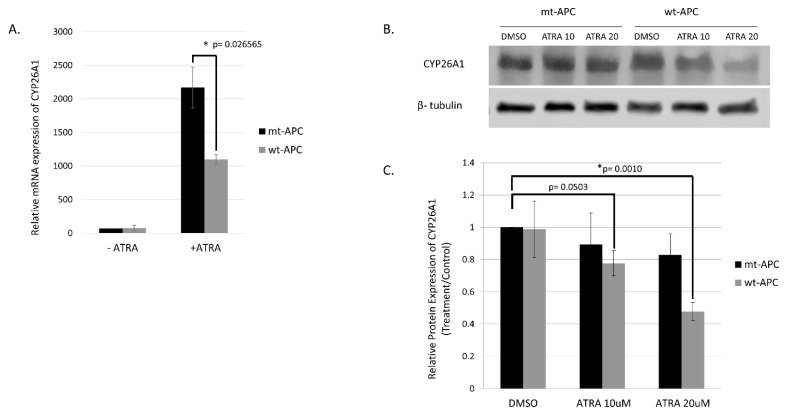
Inducing *wt-APC* expression reduces ATRA-induced *CYP26A1* expression. (**A**) NanoString mRNA profiling and western blot analysis of protein expression (**B**,**C**) revealed a 50% decrease in *CYP26A1* expression when *wt-APC* expression was induced in ATRA-treated HT29 cells. Experiments were performed in triplicate and a representative blot is shown in (**B**). Statistical significance was determined by multiple unpaired *t*-test (**A**) and one-way ANOVA (**C**). The original whole blot of Figure 5B is included in Appendix A- Western Blots.

**Figure 6 cancers-16-00264-f006:**
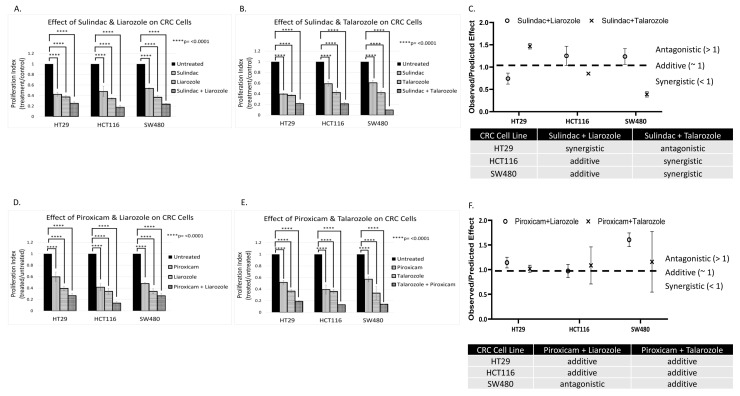
Effect of *CYP26A1* inhibitor and WNT inhibitor drug combinations on HT29, HCT116, and SW480 CRC cells. Panels (**A**,**B**,**D**,**E**) show that anti-tumor agents Sulindac and Piroxicam alone decreased the proliferation of CRC cell lines having mt-*APC*. The data also shows that *CYP26A1* inhibitors Liarozole and Talarozole alone reduced CRC proliferation. Panels (**C**,**F**) show that when Sulindac or Piroxicam was combined with Liarozole or Talarozole, the drug combination produced additive or synergistic effects. These effects were calculated as the ratio of observed response divided by predicted response (see Materials and Methods, Section 2) based on the Bliss independence dose–response model [22]. Statistical significance was determined by two-way ANOVA. Extended data showing additional concentrations are shown in Appendix A and concentrations are listed in Appendix A.

**Figure 7 cancers-16-00264-f007:**
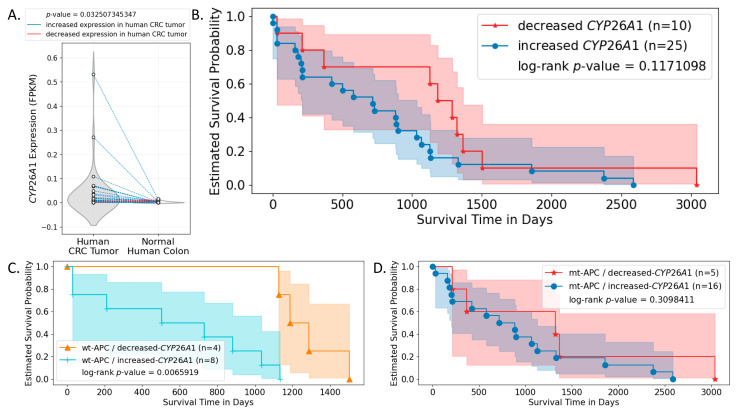
*CYP26A1* expression is a significant predictor of patient survival when patient tumors carry wild-type, but not mutant, *APC*. RNA-seq, mutation, and survival data of CRC patients were obtained from the GDC data portal and analyzed. (**A**) *CYP26A1* expression in both human CRC and normal human colon samples were plotted. Dots representing paired samples from the same patient are connected by lines. Particularly, blue lines indicate increased *CYP26A1* expression in human patient CRC samples compared to normal colon samples from the same patient; red lines indicate decreased expression of *CYP26A1* in human CRC samples compared to normal colon samples from the same patient. The shaded area indicates the 95% confidence interval. Results show that the majority of CRC patients had increased expression of *CYP26A1* in their CRC samples compared to normal colon samples, and this increase was statistically significant. (**B**) This difference in *CYP26A1* expression level between CRC and normal colon sample was not a statistically significant predictor of patient survival. (**C**) When patients were additionally grouped based on whether they carried *APC* mutations or not, results show that the change in *CYP26A1* expression between paired-tumor and normal samples was a significant predictor of patient survival only when their tumors carried *wt-APC*. (**D**) Such significance was not seen when patient tumors carried mutant *APC* (mt-*APC*).

**Figure 8 cancers-16-00264-f008:**
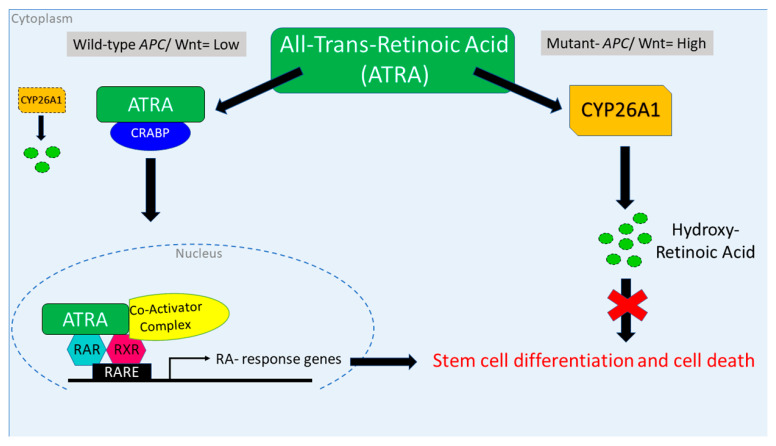
Attenuation of WNT signaling by induced *wt-APC* expression decreases *CYP26A1* and increases progression through the RA pathway. High WNT signaling levels correlate with a mutant *APC* genotype. In this case, when ATRA is administered, *CYP26A1* is overexpressed, which increases metabolism of ATRA into hydroxy retinoic acid, thereby preventing differentiation of SCs. When *wt-APC* expression is induced, components of the WNT signaling pathway, including *CYP26A1*, are not highly expressed. In this situation, when ATRA is administered, it binds to CRABP, enters the nucleus, and activates transcription of RA response genes, which induces differentiation of SCs and subsequent cell death.

## Data Availability

The source data supporting reported results can be found at Genomic Data Commons (GDC) data portal (https://portal.gdc.cancer.gov/ (accessed on 15 May 2023)), STRING database [21] (https://string-db.org/ (accessed on 1 August 2023)), the DAVID functional tool [29] (https://David.Ncifcrf.Gov/ (accessed on 27 August 2022)), and NanoString nSolver 4.0 software [20].

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
