# Peer review of "CYP26A1 Links WNT and Retinoic Acid Signaling: A Target to Differentiate ALDH+ Stem Cells in APC-Mutant CRC"

_cancers, 2024, doi:10.3390/cancers16020264_

Round 1

Reviewer 1 Report

Comments and Suggestions for Authors

The main topic of the manuscript "CYP26A1 Links WNT and Retinoic Acid Signaling: A Target to Differentiate ALDH+ Stem Cells in APC-Mutant CRC", written by Facey COB, Hunsu VO, Zhang C, Osmond B, Opdenaker LM and Boman BM. is the investigation of the effects of the crosstalk between wnt and retinoic acid signaling on the cell growth and differentiation of the colon cancer cells. In the first part the authors used cells, colon cancer and neuroendocrine cells, with wild type and mutated APC and analyzed their response on retinoic acid (RA) treatment. They also did differential analysis of pan cancer pathways panel and experimentally confirmed differential expression of several genes dependent on APC mutations.  Among them,  CYP26A1 was chosen as a possible link between wnt and RA pathways. Furthermore, colon cancer cell lines were treated with specific CYP26A1 inhibitors in combination with inhibitors of wnt signaling and the cell proliferation was analyzed. Bioinformatics' analysis of patients data was done and  CYP26A1 expression was found to be a predictor of patient survival only in patients' wild type APC tumors.

The topic is original and relevant.  Although the Introduction describes the main relations between markers and colon cancer cells, I miss more detailed explanation of the colon cancer cell biology, APC role in tumorigenesis and retinoic acid effects,  explanation of ALDH+ cells, etc.. Some data are given in the Abstract, but not in the Introduction. Also, description of abbreviations should be repeated in the Introduction. Materials and methods are well described, but more details on bioinformatics databases and tools are needed. Results are well described, but in some paragraphs  a short introduction is needed with description of the cell types (such as GLP2R+) and roles of mentioned molecules.  Also, in the Discussion, possibly more detailed relations and roles of CYP26A1 in the context of RA signaling, as well as ALDH could be described. Conclusions are consistent with the evidence. References are appropriate, and figures well presented.

Other comments

Abbreviations should be explained in the Introduction

Genes should be written in italics

line 456: sentence reorganization

Author Response

We thank the reviewers for their helpful comments. Below are each of the reviewer’s comments on our submission. Each comment (italics) is followed by our response (►).

REVIEWER 1:

Comment 1.1. Although the Introduction describes the main relations between markers and colon cancer cells, I miss more detailed explanation of the colon cancer cell biology, APC role in tumorigenesis and retinoic acid effects, explanation of ALDH+ cells, etc. Some data are given in the Abstract, but not in the Introduction.

Response ► We have expanded the Introduction to describe colon cancer cell biology, APC role in tumorigenesis, neuroendocrine cells, retinoic acid effects, and explanation of ALDH+ cells

Comment 1.2. Description of abbreviations should be repeated in the Introduction.

Response ► Thank you for the suggestion. We have added an Abbreviations section as a footnote to the Introduction to assist the reader to understand abbreviations.

Comment 1.3. Materials and methods are well described, but more details on bioinformatics databases and tools are needed.

Response ► Thank you for pointing out this omission. More details on bioinformatics databases and tools have been added to the Materials and methods.

Comment 1.4. Results are well described, but in some paragraphs a short introduction is needed with description of the cell types (such as GLP2R+) and roles of mentioned molecules.

Response Thank you for the suggestion. A paragraph has been added to the Introduction that describes the neuroendocrine cell types and the role of neuroendocrine peptides in regulating colonic stem cell populations.

Comment 1.5. In the Discussion, possibly more detailed relations and roles of CYP26A1 in the context of RA signaling, as well as ALDH could be described.

Response ► We agree. A more detailed description of the relations and roles of CYP26A1 in the context of RA signaling as well as ALDH has been added to the Discussion.

Comment 1.6. Abbreviations should be explained in the Introduction

Response ► Thank you for the suggestion. We have added an Abbreviations section as a footnote to the Introduction to assist the reader to understand abbreviations.

Comment 1.7. Genes should be written in italics

Response ► We have read through the manuscript to make sure all gene names are in italics as compared to protein names that are not italicized.

Comment 1.8. line 456: sentence reorganization

Response ► Thank you – we have rewritten the sentence.

Reviewer 2 Report

Comments and Suggestions for Authors

This is a good manuscript with good data, but there are too many abbreviations that are not explained, so the manuscript is hard to read and comprehend. You need to make the list of abbreviations and also spell out the full name of the molecule/compound when introduced for the first time.

Author Response

We thank the reviewers for their helpful comments. Below are each of the reviewer’s comments on our submission. Each comment (italics) is followed by our response (►).

REVIEWER 2:

Comment 2.1 This is a good manuscript with good data, but there are too many abbreviations that are not explained, so the manuscript is hard to read and comprehend. You need to make the list of abbreviations and also spell out the full name of the molecule/compound when introduced for the first time.

Response ► Thank you for the comments. We have added an Abbreviations section as a footnote to the Introduction to assist the reader to understand abbreviations. We have also spelled out the full name of the molecule/compound when introduced for the first time in the text.

Reviewer 3 Report

Comments and Suggestions for Authors

APC protein regulates diverse effector pathways essential for tissue homeostasis. Truncating oncogenic mutations in Apc removing its Wnt pathway and microtubule regulatory domains drives intestinal epithelia tumorigenesis. Facey and coworkers find that CYP26A1links Wnt and RA signalings using HT29 cells (referred to wt-APC and mt-APC, respectively). The manuscript is well written and CRC patients may have a chance to improve the survival. There are some concerns to publish the journal of Cancers as follows.

1)    Concerning cell differentiation, the authors show increase of neuroendocrine cell differentiation. Why is neuroendocrine cell differentiation increased and is the increase linked to Wnt and RA signaling pathways? Additionally, is the increase related to tumorigenesis?

2)    Keywords: enteroendocrine cells to neuroendocrine cells.

3)    ZnCl2 to ZnCl2 (Subscript).

4)    Page2, line69: [9]. (add comma).

5)    Materials and Methods: Sentences concerning statistical analyses should be removed. Accordingly, the authors describe in 2.9. Statistical analyses.

6)    Page3, line144-145: Please add company and model.

7)    The authors use HCT116 and SW480 cells in addition to HT29 cells. Please describe the methods for culture of these cells.

8)    2.9. Statistical analyses: p<0.05 is considered significant? p<0.01 and p<0.001 are ** and ***, respectively?

9)    Page13, line499-500: This work has vast clinical significance because it shows how CRC SCs might be sensitized to retinoid-induced cellular differentiation. The sentence seems to be overstatement.

Author Response

We thank the reviewers for their helpful comments. Below are each of the reviewer’s comments on our submission. Each comment (italics) is followed by our response (►).

REVIEWER 3:

Comment 3.1 Concerning cell differentiation, the authors show increase of neuroendocrine cell differentiation. Why is neuroendocrine cell differentiation increased and is the increase linked to Wnt and RA signaling pathways? Additionally, is the increase related to tumorigenesis?

Response ► We apologized for being unclear regarding cell differentiation. We have added a new paragraph to our revised Discussion section to provide clear explanations for the increase in neuroendocrine cell differentiation and provide answers to the questions: Why is neuroendocrine cell differentiation increased and is the increase linked to Wnt and RA signaling pathways? How is the increase related to tumorigenesis?

Comment 3.2 Keywords: enteroendocrine cells to neuroendocrine cells.

Response ► We have made this change.

Comment 3.3 ZnCl2 to ZnCl(Subscript).

Response ► We have made this change.

Comment 3.4 Page2, line69: [9]. (add comma).

Response ► We have made this change.

Comment 3.5 Materials and Methods: Sentences concerning statistical analyses should be removed. Accordingly, the authors describe in 2.9. Statistical analyses.

Response ►. We have made this change

Comment 3.6 Page3, line144-145: Please add company and model.

Response ► We have added this information

Comment 3.7 The authors use HCT116 and SW480 cells in addition to HT29 cells. Please describe the methods for culture of these cells.

Response ► We have added this information

Comment 3.8 Statistical analyses: p<0.05 is considered significant? p<0.01 and p<0.001 are ** and ***, respectively?

Response ► We have clarified that statistical significance is based on p<0.05.

Comment 3.9 Page13, line499-500: This work has vast clinical significance because it shows how CRC SCs might be sensitized to retinoid-induced cellular differentiation. The sentence seems to be overstatement.

Response ► We agree - We have toned down this statement.